# Decision Making in Networks: A Model of Awareness Raising

Federico Bizzarri [1] , Alessandro Giuliani [2] and Chiara Mocenni [1,*]

1 Department of Information Engineering and Mathematics, University of Siena, Via Roma 56, 53100 Siena, Italy
2 Environment and Health Department, Istituto Superiore di Sanità, Viale Regina Elena 299, 00161 Roma, Italy
* Correspondence: chiara.mocenni@unisi.it

**Abstract:** This work investigates how interpersonal interactions among individuals could affect the dynamics of awareness raising. Even though previous studies on mathematical models of awareness in the decision making context demonstrate how the level of awareness results from self-observation impinged by optimal decision selections and external uncertainties, an explicit accounting of interaction among individuals is missing. Here we introduce for the first time a theoretical mathematical framework to evaluate the effect on individual awareness exerted by the interaction with neighbor agents. This task is performed by embedding the single agent model into a graph and allowing different agents to interact by means of suitable coupling functions. The presence of the network allows, from a global point of view, the emergence of diffusion mechanisms for which the population tends to reach homogeneous attractors, and, among them, the one with the highest level of awareness. The structural and behavioral patterns, such as the initial levels of awareness and the relative importance the individual assigns to their own state with respect to others', may drive real actors to stress effective actions increasing individual and global awareness.

**Keywords:** awareness; self-awareness; mathematical model; network; decision making; Markov decision process; synchronization dynamics



## 1. Introduction

This work focuses on the study of how interaction among individuals can influence individuals' awareness dynamics. Throughout the work, we adopt a description of awareness as an end-stage that results from the filtering and processing of several experiences happening simultaneously in our bodies and brains [1]. This description focuses on the process and character of awareness that makes it emerge by tacit knowledge, enabling the decision maker to define (in a largely unconscious way) a sort of implicit metric space able to accommodate a given problem in a suitable class. To make a long story short, we can equate awareness to the acquisition of 'contextual' information located at a different hierarchical level with respect to the direct optimization process. Specifically, the individual applies their own policy function to make a preliminary decision, and then the outcome of their decisions is immediately perturbed any time they interact with others, due to taking into account others' awareness states. In a previous study [2], we focused on the dynamics of awareness emerging at a single individual level. However, these considerations of the individuals' awareness-raising process were not completely comprehensive, lacking an analysis of the impact exerted by the interaction of individuals. This is exactly the topic addressed with the present work, which provides an attempt to give a quantitative analysis of the crucial aspects characterizing the so-called interpersonal perspective on awareness. To accomplish this task, we adopt a network-based Markov decision model. Although purely phenomenological, the mathematical formulation gives an immediate appreciation of the effects of different styles of coping with interacting agents by the decision maker correspondent to different leadership attitudes and consensus formation.

The interest in developing such a mathematical model of awareness is manifold. From a theoretical point of view, it provides a reference for evaluating the impact of personal

relations on awareness raising, thus allowing us to analyze it from both the individual and the social perspectives [3]. Moreover, it can be used to investigate the pros and cons of using artificial intelligence in human decision making and identify new directions for integrating a prototypical consciousness into AI algorithms [4,5]. From this point of view, it is worth noting that the different individuals interacting with each other could be, in the most generic view, also non-human agents, AIs or robots for example [6]. Obviously, in these cases, the concept of awareness has a different meaning, albeit retaining both its character of 'slow accumulation' of knowledge from past examples and of the generation of 'contextual' knowledge at a different hierarchical level with respect to step-by-step optimization.

Although awareness is a very broad and multifaced concept, difficult to uniquely define, it is still possible to characterize it by extrapolating a series of peculiarities it has, which can represent the pillars for the following mathematical formulation.

**Dynamics**. Individuals' awareness is a dynamic process [7], continuously evolving over the individual's lifetime, which can be mathematically represented by a sequence of states. **Directionality**. Awareness represents a certain state of the individual, which completely incorporates their past, and its evolution results from a conscious directed focus, motivation, and effort of the individual toward the increase of their level of awareness [8]. **Optimization**. This goal-directed behavior, natural in human lives, can be mathematically transposed by an optimization process, depending on the state itself, which tends to obtain the maximum benefit. This benefit is assumed, in our case, as directly proportional to the state, which in turn indicates the level of awareness, so that living with a higher level of awareness produces higher individual wellbeing from the physical, psychological, and emotional points of view.

**Decision making.** From the general considerations reported above, we particularly focus on the relation with the decision making process, claiming that the state of awareness determines the choices the individual made. **Uncertainty**. Choices are anyway subjected to external uncertainty given by uncontrollable environmental factors. **Reasoning propensity.** Each individual is characterized by a certain reasoning propensity which embodies the preferred way to process the available information about the decisional problem to face, and results from a combination of an *intuitive* and an *analytical* component, as proposed in [9]. For the purpose of our study, the intuitive individual thinks to have access to personal abilities, not related to cognition, allowing their level of awareness to increase using an appropriate degree of intuition; this has to do with the personal confidence in tacit knowledge. Vice versa, the analytical individual faces the efforts to acquire and elaborate quantitative information they use to make their decisions. **Information costs.** The knowledge of their propensity determines, for the individual, a reduction of the incurred costs defined as the resource's consumption deriving from the process of acquiring, processing, and analyzing all the data about the problem. This knowledge represents the *intrapersonal* dimension of self-awareness reported in the definition given by Carden et al. [7], already explored in our previous work [2], opposite but also integrated by the interpersonal dimension, upon which is focused the present work. **Backward induction.** Moreover, the backward induction mechanism applied to solve the optimization process explains not only the future choices, but also the fact that naturally the individuals tend to reconstruct their past choices, and consequent outcomes in terms of wellbeing, a posteriori, as if we imagine solving the decision problem from the present state in a past horizon.

The reported characteristics and mechanisms can be modeled by the sequential model class of Markov Decision Processes (MDPs), mathematically formulated in the next section. However, this is not enough, and an additional perspective must be considered in order to take into account the interaction component among individuals.

In fact, interaction is an essential component of human life, an aspect from which one cannot escape. It is said that "people seek the truth collectively, not individually" [10]. This collective dimension implies that the decision process, in many cases, becomes an emergent property of a given set of individuals. To mathematically shape the interaction

among a given set of individuals and to analyze the resulting behaviors and dynamics, it is possible to formalize it in terms of networks in which the nodes are different individuals and the edges their pairwise interaction [11]. In our approach, we formalized the effect of interaction as impinging on the single individual decision process and thus acting as a modifier of their dynamics.

As mentioned before, the concept of awareness synthesizes two different but equally important perspectives: the first corresponds to an intrapersonal aspect centered on the awareness of one's own resources and internal state, as well as any kind of knowledge that the Decision Maker (DM) can learn from the context. The second, defined as interpersonal, focuses on the impact exerted by other agents on the subject and vice versa [7] and is equally important and essential as the first. The interpersonal perspective can be further separated into two components: (a) Behavior corresponds to an externally visible action that might directly affect others; (b) Others' perception corresponds to how an individual is perceived by their neighborhoods and has to do with the ability to "receive and interpret" feedback.

These components join the intrapersonal perspective, requiring the individual to be consciously aware of and to understand how their decisions could influence and be influenced by the other's choices. Sometimes awareness and self-awareness can be traced back to the interpersonal view defining it as "anticipating how others perceive you, evaluating yourself and your actions according to collective beliefs and values, and caring about how others evaluate you" [12]. Other times, the literature emphasized both components, claiming that awareness consists of the internal—recognizing one's own inner state—and the external—recognizing one's impact on others [13], but the second component is no more elaborate than pointing out that it was a self-evaluative component referring to becoming conscious of one's impact on others. The interpersonal perspective is the focal point of this work.

A relevant example case showing the importance of the interpersonal aspect comes from the theme of Leadership. Although scholars continue to debate how to define, detect, and measure awareness in people, most of them would intuitively regard it as fundamental to effective leadership behavior [14]. For example, it has been established that by becoming more self-aware, leaders better understand their strengths and self-resources and better identify where they might need further development [15].

In our opinion, these aspects of intrapersonal perspective should be complemented by a second, sometimes neglected, interpersonal view. This is critical to leadership and corresponds to the ability to interpret (1) the emotions, thoughts, and preferences of others and (2) the influence leaders have on people. Because leadership is relational [16], leaders must develop peer-to-peer connections with their team members, so they become aware of their perspectives, hopes, goals, strengths, and developmental needs and achieve the full extent of the desired outcome and the ability to anticipate their impact and influence on others.

The manager who is consistently inclined to consider only profits or goal attainment, rather than others' perceptions, as well as the general, who is only focused on winning the battle, rather than being constantly connected to their subordinates, may restrict the ability to be an aware leader [14]. If leaders are not aware of their impact on others and if they cannot effectively diagnose how others experience their efforts, then their ability to lead effectively will be severely hampered. To cope with these limitations, leaders should develop a greater appreciation for extrinsic standards, such as an awareness of how others perceive them as well as of their own emotions and the emotions of others, thus building an overall positive work environment.

The case of leadership is an example highlighting the importance that the interpersonal dimension covers in many different fields of our lives. Extending the concepts reported above to the framework proposed in this paper, aimed at theoretically describing the impact of others' behavior on individual awareness, we will sketch a prototypical model

based on interaction networks in which the 'neighborhoods sentiment' is modeled (from the individual's point of view) as a global network response generated from its wiring structure.

## 2. Methods

This section reports a mathematical formulation that can be considered a first attempt to evaluate how the interaction among individuals impacts the dynamics of awareness in the context of decision making processes. The starting point is a framework embedding the different phenomena impacting the awareness-raising process, with a focus on the single individual in isolation. Then, a reasonable formulation is introduced to describe the contribution on the dynamic of the single agent provided by the interactions among others, exploiting a network and a suitable coupling function.

### 2.1. Modeling Individual Awareness

To provide a mathematical model embedding the basic mechanisms of awareness raising, we start from the wide definition of a deterministic finite-state automaton as a quintuple ($\Sigma$, $X$, $x_0$, $\theta$, $F$) where $\Sigma$ is the input alphabet, consisting of a finite non-empty set of symbols, $X$ is a finite non-empty set of states, $x_0 \in X$ is an initial state, $\theta : X \times \Sigma \to X$ is a transition function, and $F \subseteq X$ is the set of final states. Then we focus on the framework of Markov Decision Processes (MDPs) which embed a Markov-chain with the addition of input ($u_t$) and costs/rewards, according to a function $r$ [17,18].

The MDP is then defined as a tuple ($S$, $U$, $P$, $r$) where $S$ is a finite set of feasible states, called state space, $U$ is a finite section of available actions, namely the action space, $P : S \times S \times U \to [0, 1]$ is a stochastic transition function defining the probability to shift from a state $s$ to a state $s'$ by choosing the input $u$, and $r : S \times S \times U \to \mathbb{R}$ is a reward function incurred.

A mathematical model grounded on MDPs describing the basic elements which impact the individual's reasoning and analyzing how they are related to the process of increasing personal awareness is shown in Figure 1, which reports a graphical scheme of the mathematical model where the individual mechanisms introduced in [2,19] (green border sub-scheme), and the networked system of interconnected individuals (complete scheme) are highlighted.

In particular, the single individual model considers a discrete and finite time horizon of length $T$, in which each time-epoch, $t$, corresponds to a moment of making a decision, $u_t \in U = [0, 1]$, under the conventional assumption that the more analytical the choice, the higher the value of $u_t$, and, vice versa, the more intuitive the choice, the lower the value of $u_t$. The state, $s_t \in S = [0, 1]$, of the individual at each time $t$, is a representation of their level of awareness, incorporating all past experiences. Therefore, awareness is considered to be a process—mathematically a sequence of states—involving the DM's experiences, filtered by their perspectives, beliefs, and values. Moreover, in a MDP we have at the same time the presence of a choice of the DM and uncertainty about its outcomes given by uncontrollable external factors, represented in our framework by a *stochastic variable* $w_t \in W = \{1, 0, -1\}$, as always happens in our decisions. This model is good trade-off between realism and simplicity: broad enough to account for realistic sequential decision making problems while simple enough to allow it to be understood and applied by different kind of practitioners. Each individual possesses a reasoning propensity, $p_r \in [0, 1]$, which embeds the specific attitude in processing the information about the problem and represents the trade-off between two diametrically opposed reasoning modalities: intuitive ($p_r = 0$) and analytical ($p_r = 1$), assuming in this way, that both are always involved, with different amounts, in any decision [20,21]. The reasoning propensity affects the *policy*, $\mu$, of the individual. A policy is a function prescribing the DM the action to take for each possible state at any time instant, and it is represented by a matrix of dimension $|S| \times T$. Therefore, the policy turns out in the decision:

$$u_t = \mu(s_t, t) \; \forall \; s_t \in S, \; t = 1, \ldots, T. \tag{1}$$

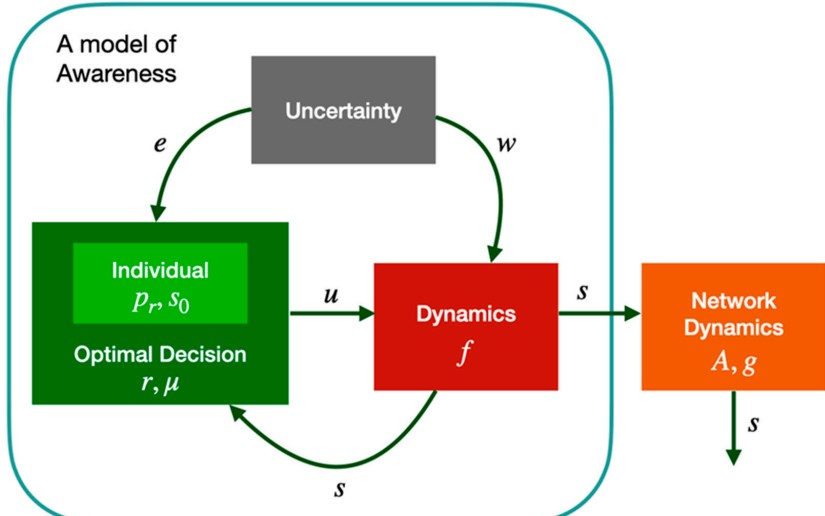

**Figure 1.** Pictorial representation of the model of awareness. The main elements of the mathematical model are: 1. The individual characteristics (light green box) consisting of the awareness state $s$, the reasoning propensity $p_r$, and the initial level of awareness $s_0$; 2. The optimization process (dark green box) defined by the reward function $r$, and the optimal policy $\mu$; 3. The awareness dynamics (red box) determined by the action $u$, the sources of uncertainties $e$ and $w$ (gray box), and the transition function $f$; 4. The effect of the interconnection with others (orange box), where $A$ is the adjacency matrix of the graph and $g$ is the interaction function.

The choice, driven by the policy as exposed in Equation (1), leads to two results: the DM receives a reward, defined by a reward function $r(s_t, u_t, w_t)$, and the system evolves to a possibly new state according to a stochastic transition function $f$ defined as:

$$s_{t+1} = f(s_t, u_t, w_t) \tag{2}$$

Equation (2) characterizes the dynamics of the process of self-knowledge indicating how the future level of awareness of the individual depends on the current state, the choices they make, and its outcome which is subject to some uncertainty represented by $w_t$. We assume, for simplicity, that by deciding $u_t$, the state $s_t$ can increase ($s_{t+1} = s_t + 1$, when $w_t = 1$), decrease ($s_{t+1} = s_t - 1$, when $w_t = -1$), or remain the same ($s_{t+1} = s_t$, when $w_t = 0$), according to a certain probability expressed by the transition probability function $P$. In particular, we define a stationary probability $P^S(u_t)$ constant for all $u_t$, a forward probability $P^F(u_t)$ as a function of $u_t$, and a backward probability $P^B(u_t)$ computed starting from the first two: since as probabilities their sum must be equal to one. The presence of uncertainty affecting the outcomes of the decision, given by uncontrollable elements in the environment, makes the state evolution and the rewards sequence stochastic. The reward function incorporates the assumption that the wellbeing increases with the state, which in turn indicates the level of awareness, so that living with a higher level of awareness produces a higher individual's global wellbeing. On the other hand, it introduces a component related to information costs, which are higher the more the choice of the individual is analytical, i.e., in our convention, the more $u_t$ is near 1 (see Appendix A for more details on the reward function $r$).

The individual is focused on the maximization of their wellbeing i.e., the sequence of rewards they incur over time, embedding a directionality typical of the human goal-directed behaviors. The optimal decisions are obtained by maximizing the sequence of expected rewards, according to the following problem:

$$\max_{\mu} E\left[\sum_{t=0}^{T-1} r(s_t, \mu(s_t, t), w_t) + r(s_T)\right],$$
$$s.t. \ s_{t+1} = f(s_t, \mu(s_t, t), w_t), \ t = 1, \ \dots \ , T. \tag{3}$$

The last reward the DM incurs at the last time-epoch $T$, namely the term $r(s_T)$ in Equation (3), *is* a priori fixed because it is used as the starting point for the reconstruction of the optimal policy in the process of optimization described below, embedded applying a backward induction method. We assume that the terminal benefits the DM incurs at the final time $T$ increases by increasing the values of the final state $s_T$. Moreover, a factor $\delta$, weighting future rewards, has been introduced.

The maximization problem expressed in Equation (3) is recursively solved by implementing a Dynamic Programming algorithm, where the original problem is separated into a recursive series of easier sub-problems considering a shorter time horizon from $\tau$ to $T$ and given the step-by-step initial state $\underline{s}$, representing the state at time $\tau$. The expected reward function to optimize at each stage is explicitly described as:

$$V_\tau(\underline{s}) = r(\underline{s},\ \mu(\underline{s},\tau)) + \delta\left[\gamma_1 V(\underline{s}+1)P^F(\mu(\underline{s},\tau)) + \gamma_2 V(\underline{s})P^S(\mu(\underline{s},\tau)) + \gamma_3 V(\underline{s}-1)P^B(\mu(\underline{s},\tau))\right], \qquad (4)$$

where the coefficients $\gamma_1$, $\gamma_2$ and $\gamma_3$ weight the cases of increasing, maintaining constant, or decreasing the state. These parameters allow us to take into account different attitudes of the DM, for example penalizing the eventuality of decreasing awareness.

The optimization process, intended to maximize the sequence of rewards expressed in Equation (4), embeds in itself an element of individual's self-observation, which determined the action/decision to made. This feedback component allows the DM to build a policy by having knowledge of the form of the transition probability function, and the current state at each time epoch. In this way, the DM can overcome their usual, automatic process by shifting from their habitual to a new aware policy, which results from the reward maximization process, thus mitigating the mechanistic tendencies of the individual [2]. Because of the dependence of the reward on the level of awareness, this maximization also models the fact that the increase of personal awareness follows a focus on this specific objective, emerging from personal effort and motivation [22].

It is possible to include in the model the individual's immediate emotions [23–25], considering also that the role played by emotions could depend on the present level of awareness of the individual. For example, at a low level of awareness, emotions prevail on individual reasoning, so DM is driven in its choices by the research of instant satisfaction. In this condition, the future (intended to be long-term perspectives) will have very little weight on their choice, the DM is enslaved by immediate reward, a condition which could turn out to be highly damaging. This 'immediate reward' dynamics loses its importance when the individual reaches a high level of awareness. In the mathematical model, emotions affect the function $\delta$, showing how the individual differently weighs the future with respect to the present and taking into account also the dependence of function $\delta(s)$ on the state $s$.

Once the generic structure of the model has been defined, it is possible to assign specific forms to the different model's functions to perform numerical simulations aimed to study the emergence of behaviors and dynamics. The different functions used are reported in Appendix A.

Figure 2 reports a comparison of the aware policy with respect to a habitual policy referred to as the most basic and simplest mechanism governing an individual's usual decisions. It assumes that decisions spring from habits, automatic, non-conscious mechanisms, represented in the model by individual reasoning propensity $p_r$. Furthermore, it is fixed and constant in time, with the only addition of noise $e_t$, as reported in Equation (5), normally distributed with mean zeros and variance $\sigma$ (where $\sigma$ is fixed to 0.08 in the simulations). It takes into account a source of uncertainty, mainly due to external factors, influencing the effective choice and shifting it around the reasoning propensity of the individual.

$$\mu(s_t, t) = p_r + e_t \ \forall s_t \in S,\ t = 1, \dots, T. \qquad (5)$$

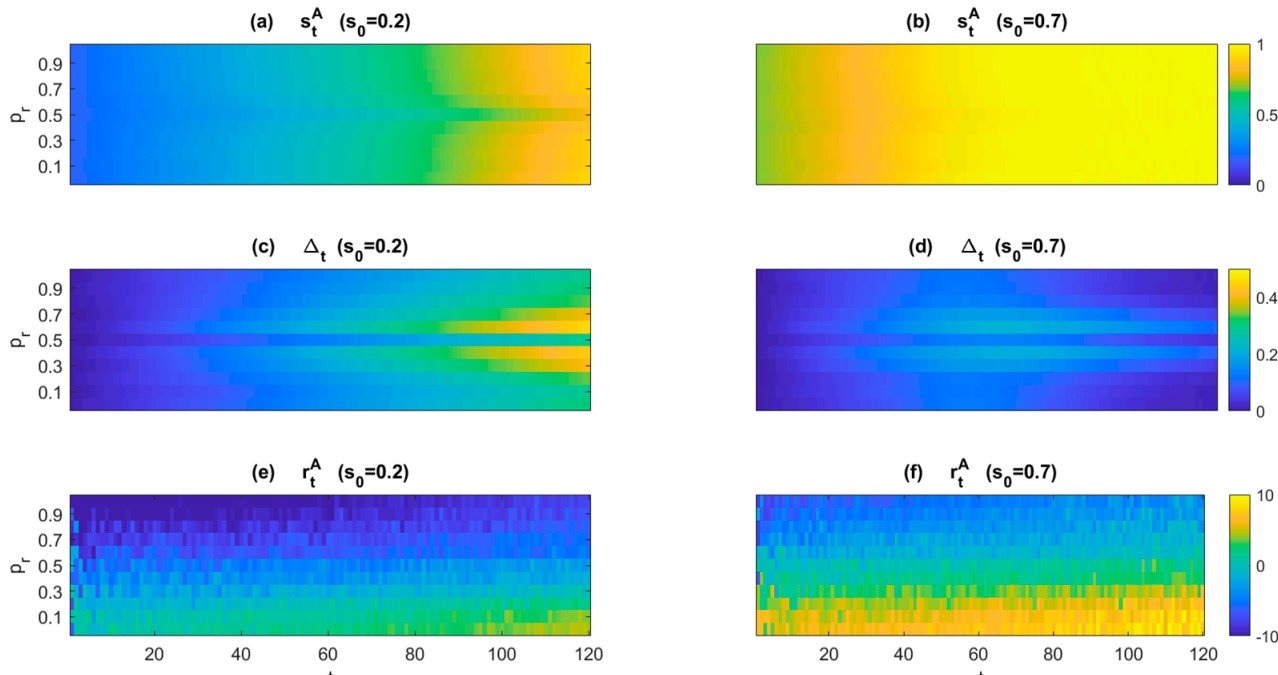

**Figure 2.** Comparison between aware (A) and habitual (H) states and rewards over time, for individuals with different levels of reasoning propensities $p_r$. The color bar reports the different variables of interest. For the two upper panels (**a**,**b**) the color bar indicates the level of the state, for the two middle panels (**c**,**d**) the payoff obtained by applying an aware policy with respect to the habitual one, where $\Delta_t = s_t^A - s_t^H$, and in the two bottom panels (**e**,**f**) the reward for an aware policy. All the panels on the left consider a low initial state ($s_0 = 0.2$) while the ones on the right consider a high initial level ($s_0 = 0.7$). The aware model's parameters used for the comparison are fixed to $\alpha = 1$, $\beta = 1.5$, $\gamma = [5, 1, 0.01]$, and $\delta = 0.75$. The functions used to perform the simulations are described in detail in Appendix A.

On the contrary, the aware policy is embodied by the faceted optimization process described in Equations (3) and (4).

Figure 2 shows, before anything else, the effect resulting from the application of the aware policy in terms of a high final level of the state (near 1) starting from a low state (Figure 2a) and speed in reaching the maximum state level starting from a high initial state (Figure 2b). The shift from a habitual to an aware policy has an improving action on the dynamics of the state (Figure 2c,d), except when the initial level of awareness is high enough (see Figure 2d, where $s_0 = 0.7$) and after a long time horizon (the similarity at the first time instant in Figure 2c,d is due to the equal initial state assumptions). The benefit of the shifting increases as the $p_r$ grows from zero to 0.5 where there is a discontinuity for which the difference between the two situations is neglectable, and then symmetrically decreases as the $p_r$ grows further from 0.6 to 1. This fact is due to the symmetrical form assumed for the theoretical probability functions defined for an analytical and intuitive individual, as expressed in Equations (8) and (A1), respectively. In the end, the two panels **e** and **f** highlight how the improvement of the state dynamics not necessarily corresponds to an increase in the reward, $r_t$, because the model considers also a part of costs related to the level of analyticity adopted in the decision so that an individual can reach a high level of the state but can also have some losses simply indicating a high cost due to excessive use of analytic reasoning.

### 2.2. Interactions and Awareness: A Network Model

The framework exposed until now describes the emergence of awareness considering the decision making process of a single agent taken into isolation. However, it lacks an

essential dimension: how the interaction with other agents can impact the awareness dynamic of every single agent. The novelty of the present work is specifically to establish a mathematical declination of the interactions among agents in the optic to study the awareness-raising phenomenon. This interaction can be modeled through a Network—also called a Graph in the mathematical literature—that is as a collection of vertices—*nodes*—joined by links—edges [26,27]. Whether time series of behavioral data are available for each node, the links can derive from pairs' partial correlation coefficients, mutual entropy, and so on. Furthermore, conditional probabilities computed over some relevant variable or distance between pairs of nodes over a metric space can be used as quantifiers of edge strength [28]. Since the model developed in this study is based on theoretical assumptions as well as behavioral and psychological evidence of prototypical cases, the connections are undirected and weighted uniformly in the population. Estimating specific values for these edges is crucial; to this aim, suitable surveys are under development by the authors of the present paper.

According to the common notation applied in the mathematical literature, a graph is defined by giving the set of $n$ nodes and the set of $m$ edges linking them. The effective meaning of nodes and edges depends on the specific context in which the graph is applied: in our case, the nodes are the agents [29], and the edges model a relationship among them, in the sense that each agent is influenced directly by their neighborhoods, i.e., all the other agents it is connected to, and indirectly by all agents without any specific dependence on the distance. Now, there are many ways to represent a network mathematically, but typically it is given the adjacency matrix $A$ in which each element $a_{ij}$ indicates the presence or absence of a link between nodes $i$ and $j$, and eventually the weight of that connection [26].

The kind of network we are now considering has at maximum one edge between the same pair of vertices, in opposition to a multi-edge graph; moreover, there are no edges that connect vertices to themselves, the so-called self-edges or self-loops. A network of this type that has neither self-edges nor multi-edges is called a simple network or simple graph. The network considered is undirected and unweighted, so the adjacency matrix has all zeros on the diagonal (no self-loop) and it is symmetric. All $a_{ij}$ are equal to 1 or 0 to indicate the presence and absence, respectively, of an interaction between nodes $i$ and $j$; they represent simple on/off connections among different agents.

As future development, it might be interesting to consider a weighted graph by explicitly defining a certain strength/weight of each edge connecting nodes $i$ and $j$, represented by $a_{ij}$. It might be equally interesting to consider a direct graph for which the adjacency matrix is not symmetrical, exploiting the fact that a connection between $i$ and $j$ does not have the same weight in both directions.

The aim of this work is to evaluate the mutual impact of individuals' decisions on their respective awareness dynamics. To mathematically analyze this phenomenon, it is necessary to introduce a dependence among individuals characterizing their interactions [30,31]. A possible way to insert this interaction is to define the state evolution of each agent as related to the state of the other agents in the network. From a mathematical point of view, we introduce a new interaction term $g_{ij}\left(s_t^j, s_t^i\right)$, claiming that the state evolution for each agent is governed by the function:

$$s_{t+1}^i = s_t^i + \frac{1}{k_i} \sum_j a_{ij} g_{ij}\left(s_t^j, s_t^i\right) + w_t^i \tag{6}$$

For the interaction term, we can assume that there exists an imitative behavior driving people to behave like their neighbors. The same consideration is applied in scientific literature to describe mechanism of "gossip", in which people tend to copy their friends: they increase the amount they talk about a topic if their friends are talking about it more than they are, and decrease if their friends are talking about it less [26].

This imitative behavior can be represented by putting:

$$g_{ij}\left(s_t^i, s_t^j\right) = g\left(s_t^j\right) - g\left(s_t^i\right). \tag{7}$$

This is the form introduced for the first time to describe the synchronizing Kuramoto oscillators [32,33].

If we consider, for Equation (7), the simple linear case for which $g(s_t^j) = \varepsilon_1 s_t^j$ and $g\left(s_t^i\right) = \varepsilon_2 s_t^i$, then Equation (6) reads as follows:

$$s_{t+1}^i = s_t^i + \frac{1}{k_i} \sum_j a_{ij}\left(\varepsilon_1 s_t^j - \varepsilon_2 s_t^i\right) + w_t^i \tag{8}$$

Indicating that the state evolution of the agent $i$ depends on its previous state ($s_t^i$), on the stochastic process, $w_t^i$, and on an interaction term that describes the dynamics of all the nodes $j$ linked to node $i$. This term is the summation of all the differences between the state values of the considered node and the ones of the nodes to which it is connected, tuned using two constant positive parameters $\varepsilon_1$ and $\varepsilon_2$. These parameters reflect how the present state of the individual is weighted with respect to the ones of the neighbors (considering, in this first case, an equal coefficient for all the neighborhoods of node $i$). When $\varepsilon_1 = \varepsilon_2 = 0$ the interaction term present in Equation (8) is null, and the state evolves as the single individual awareness model. In the end, the summation is modulated with a term $\frac{1}{k_i}$, where $k_i$ is the degree of node $i$ (i.e., the number of its neighborhoods).

Recalling that the state value $s_t^i$ is defined as the level of awareness of agent $i$ at time $t$, we can appreciate the meaning of the proposed equation. The form of the equation encompasses the fact that neighborhoods of $i$ with higher state values have a positive impact on the dynamic of node $i$, while neighborhoods with lower values have a negative impact on it, i.e., agents with a higher and lower level of awareness can help or damage, respectively, the level of awareness of the node $i$.

### 3. Results

By applying the mathematical framework described above, it was possible to evaluate the resulting dynamics by solving the optimization problem defined in Equation (3) and analyzing the corresponding simulations.

In these simulations, we applied the aware policy described above also considering the presence of immediate emotions. In our previous work [2], we analyzed how emotions, when they embed a dependence on the level of awareness, could have a very different impact on individual's dynamics depending on the specific level, so they can speed up the dynamic when the individual has a high state of awareness, whereas can be more damaging the more the state is low. It is possible to observe this phenomenon in Figure 3c,d, considering the dynamics without interaction and showing that individuals with an initial state under a certain threshold (around 0.8) suffer from a negative impact of emotions that prevents the state to increase.

Moreover, Figure 3e,f highlight a synchronization mechanism that involves the individuals and diffuses on the network, eventually becoming more evident at increasing number of nodes. This process corresponds to the information spreading through the network making all the individuals to reach equal values of the ending state. The collapsing to a single attractor corresponding to a uniform state of all the nodes, is a consequence of the high level of correlation between nodes, which is precisely a universal of tipping points before a phase transition [34–36]. The correlation effect (and the consequent generation on unanimity) is extended by the recurrence condition allowing each individual to have a 'complete view' of the entire system. Because the network is represented by a connected graph, each individual has an awareness of the entire system: the state depends on the state of the neighbors, but in turn their states depend on the states of their neighbors and so on. Therefore, ultimately, the state of each node depends on the state of all other nodes,

be they closer (in the neighborhood) or further away, at the margins of the system. Given that the decision of the individual derives from the policy, which has a specific value for each possible state at each time instant, the coupling term produces an impact on the state dynamics but also on the choices. The policy remains the same, it is not influenced by the network, in the sense that the single individual has their own way of reasoning, but the effective values suggested by the policy, depending on the state evolution, are affected by the presence of interaction.

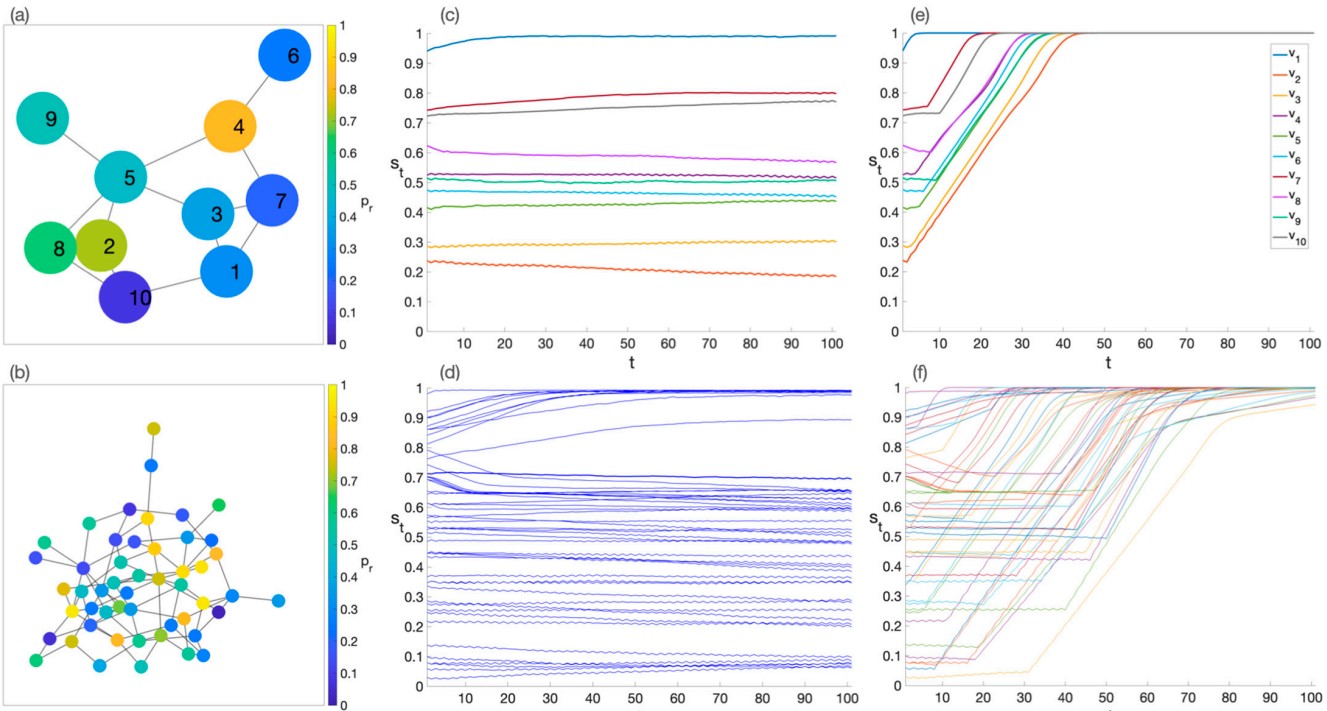

**Figure 3.** State evolution in a network of 10 and 50 individuals. Panels (**a**,**b**): in both the 10 and the 50 nodes cases, the graph topology is Erdős–Rényi with an average degree of 4. The colors indicate the value of the reasoning propensity ($p_r$) for all nodes, uniformly distributed in the open interval (0, 1). The same is done for the initial state, which is uniformly distributed in the open interval (0, 1). State dynamics $s_t$ without panels (**c**,**d**) and with panels (**e**,**f**) interactions among individuals. The synchronization parameters are $\varepsilon_1 = 0.05$ and $\varepsilon_2 = 0.01$ and there is included the presence of emotions. All the other parameters are $\alpha = 1$, $\beta = 1.5$, $\gamma = [5, 1, 0.01]$, and $\delta = 0.75$.

It is interesting to notice that over the network, the state overcomes the stopping phenomena reported above (Figure 3c,d), enabling all the individuals to asymptotically reach the maximum level of the state ($s_t = 1$), independently from their initial state (Figure 3e,f). Table 1 reports simple statistical metrics based on the mean and variance of the state across the 10 and 50 nodes network models with and without interaction. A rapid increase in mean awareness level, coupled with a progressive synchronization (decrease in variance) of different agent states, can be observed. Moreover, it is worth noticing that the lack of change in time in the 'without interaction' condition stems from the much longer characteristic time of isolated nodes' awareness increase with respect to the interaction condition.

Different combinations of the coefficients $\varepsilon_1$ and $\varepsilon_2$, which weigh the interactions, have been analyzed (see Figure 4a), highlighting the different pairs that can provide a worsening or an improvement of the state evolution. We note how similar parameters' values have very different effects, defining a diagonal of the matrix; pairs with helpful influences ($\varepsilon_1 > \varepsilon_2$) are positioned above the diagonal, whereas pairs with harmful influences ($\varepsilon_1 < \varepsilon_2$) are below it.

**Table 1.** Statistical comparison of the model simulations developed for the two networks of Figure 2. The mean and the variance of the simulations without (Figure 3c,d) and with (Figure 3e,f) interactions are evaluated at equally spaced time instants, from the initial time $t_0$ to the final time $T$. The values reported in the table are approximated to two digits.

| Time | Metric | Network with 10 Nodes | | Network with 50 Nodes | |
|---|---|---|---|---|---|
| | Mean/ Variance | Without interaction | With interaction | Without interaction | With interaction |
| $t = t_0$ | $\mu$ | 0.6 | 0.6 | 0.5 | 0.5 |
| | $\sigma^2$ | 0.05 | 0.05 | 0.09 | 0.09 |
| $t = T/4$ | $\mu$ | 0.6 | 0.9 | 0.5 | 0.5 |
| | $\sigma^2$ | 0.05 | 0.02 | 0.1 | 0.1 |
| $t = T/2$ | $\mu$ | 0.6 | 1 | 0.5 | 0.7 |
| | $\sigma^2$ | 0.06 | 0 | 0.1 | 0.06 |
| $t = 3T/4$ | $\mu$ | 0.6 | 1 | 0.5 | 0.9 |
| | $\sigma^2$ | 0.06 | 0 | 0.1 | 0.02 |
| $t = T$ | $\mu$ | 0.6 | 1 | 0.5 | 1 |
| | $\sigma^2$ | 0.06 | 0 | 0.1 | 0 |

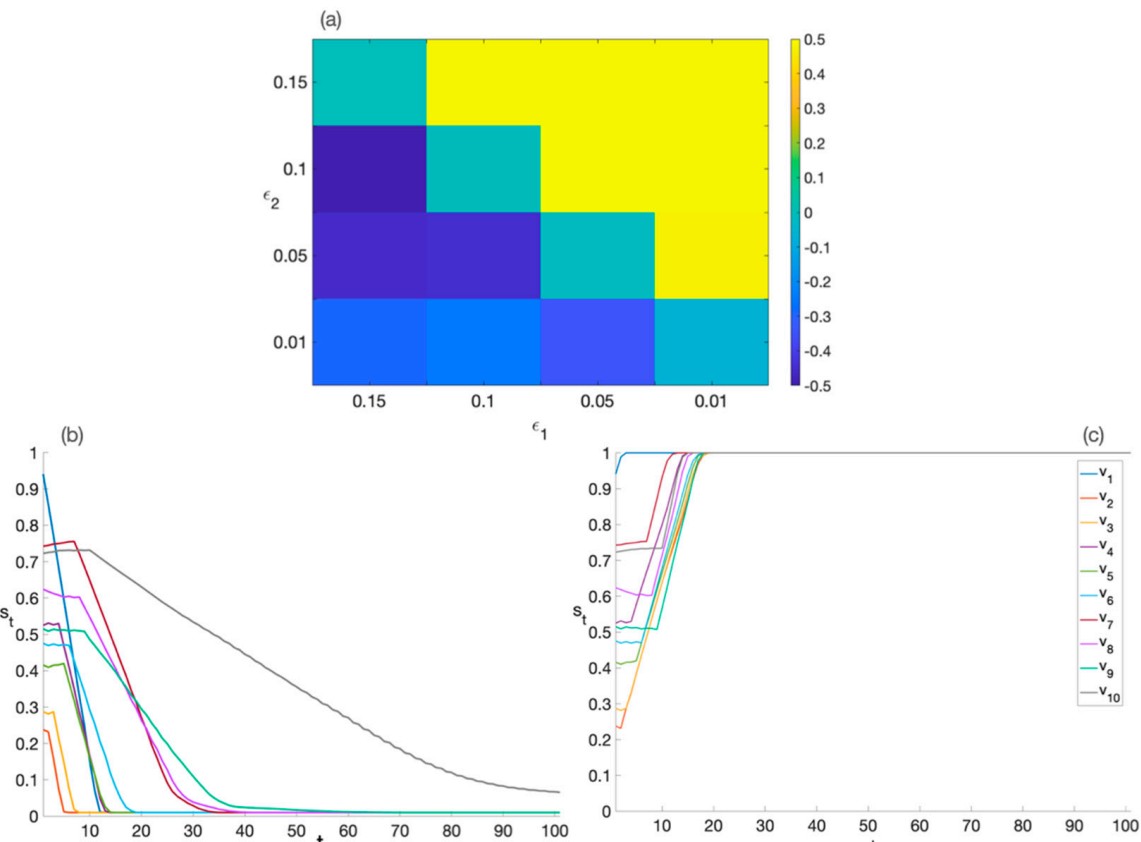

**Figure 4.** Comparison of pairs of state evolutions, with and without interactions, for different combinations of the parameters $\varepsilon_1$ and $\varepsilon_2$ (panel **a**). The color indicates the gain obtained considering each specific pair. It is computed as $\frac{1}{n} \sum_{i=1}^{n} (\int s_t^i - z_t^i dt)$, where $s_t^i$ and $z_t^i$ are the state evolutions with and without interaction, respectively, and $n$ is the number of nodes. State evolutions for harmful (panel **b**, $\varepsilon_1 = 0.01$, $\varepsilon_2 = 0.1$) and helpful (panel **c**, $\varepsilon_1 = 0.1$, $\varepsilon_2 = 0.01$) interactions, respectively. All the other parameters are $\alpha = 1$, $\beta = 1.5$, $\gamma = [5, 1, 0.01]$, and $\delta = 0.75$.

Moreover, in Figure 4 two extreme cases are reported: the first (Figure 4b) considers a pair which makes interaction harmful, and the second (Figure 4c) which provides a helpful result.

## 4. Discussion and Conclusions

The aim of this research was to propose a model characterizing the dynamic of awareness raising in individual's decision making embedding an interpersonal perspective, to take into account the presence of a bi-directional influence of other agents. To do this we considered a simple and immediately comprehensible coupling term, claiming that the state evolution of each individual depends on the state of all their neighbors. This term incorporates in the state evolution of each node $i$ a factor $(\varepsilon_1 s_t^j - \varepsilon_2 s_t^i)$ which depending on the distance between its state and the one of all its neighbors, weighted by two constant factors. This difference claims that for each node:

- Neighbors with higher state level are helpful in the state evolution.
- Neighbors with lower state level are harmful in the state evolution.

Since the considered network is fully connected, each node is influenced directly or indirectly by the states of all the other nodes in the graph.

For particular combinations of $\varepsilon_1$ and $\varepsilon_2$, a synchronization mechanism is observed among the nodes of the network allowing each individual to reach the highest level of the state. This is an interesting way to overcome the problems related to the presence of emotions, indeed, for individuals starting from low states of awareness, it enables a further increase of the state.

Different coupling mechanisms can be considered, also taking into account more complex relationships among individuals, such as weighted and directed graphs, as well as different kinds of random graph topologies, such as scale-free degree distributions that have been recognized as ubiquitous in social relations.

In this work, the focus is on personal choices made with an explicit intention/motivation of increasing personal awareness, consequently each decision maker has their own way of reasoning that cannot be modified, but surely influenced, by the other individuals. It is said that "people seek the truth collectively, not individually" [10] and also the awareness-raising process must take into account this collective perspective. It is essential to enrich the personal self-awareness with the inclusion of a knowledge of the external environment, for example considering the state of all other individuals to which a decision maker is inevitably related in their decisions. It has been analyzed how this impact can provide also beneficial stimulations, allowing the individual to overcome personal phenomena preventing further raising of awareness.

To the best of our knowledge, this is the first explicit quantitative model addressing the awareness dynamics in an interpersonal context, this generates the obvious limitation of the impossibility of a comparison with alternative formulations and, still more important, with real-life situations. As for this second point, we are planning a benchmark study with suitable real-life data. The relative data sources should be carefully chosen in order to obtain reliable proxies of collective awareness raising: we can speculate the ideal situation could be a training process as in introductory sport, dance, or music schools. The benefits of the proposed model are its complete transparency and analytical character allowing for a full explanation of the obtained results.

**Author Contributions:** Conceptualization, F.B., A.G. and C.M.; methodology and formal analysis, C.M. and F.B.; software, F.B.; validation, F.B., A.G. and C.M.; writing—original draft preparation, F.B.; writing—review and editing, C.M. and A.G. All authors have read and agreed to the published version of the manuscript.

**Funding:** This research received no external funding.

**Institutional Review Board Statement:** Not applicable.

**Informed Consent Statement:** Not applicable.

**Data Availability Statement:** No real data have been used for this study. The Matlab code developed for the simulations can be made available upon request.

**Acknowledgments:** F.B. and C.M. thank Jean Carlo Pech de Moraes for the fruitful meetings, suggestions and for his encouragement to proceed with this study.

**Conflicts of Interest:** The authors declare no conflict of interest.

## Appendix A. Model Functions and Parameters Specification

This appendix reports the specific functions' shapes applied for developing the model simulations reported in the paper.

$$r(s_t, u_t, w_t) = \alpha s_t - \beta u_t, \text{ with } \alpha, \ \beta > 0, \tag{A1}$$

$$r(s_T) = e^{s_T}, \tag{A2}$$

$$f(s_t, u_t, w_t) = s_t + w_t, \tag{A3}$$

$$P^A(u_t) = \frac{(\overline{u}_t - a) + b}{(\overline{u}_t - a)^2 + c} + d, \tag{A4}$$

$$P^I(u_t) = \frac{(\overline{u}_t) + b}{(\overline{u}_t)^2 + c} + d, \tag{A5}$$

$$P^F(u_t) = p_r P^A(u_t) + (1 - p_r) P^I(u_t), \tag{A6}$$

$$P^S(u_t) = k, \tag{A7}$$

$$P^B(u_t) = 1 - \left( P^F(u_t, 1) + P^S(u_t, 2) \right). \tag{A8}$$

Equations (A1) and (A2) describe the reward functions at a generic time $t$ and at the final time $T$, respectively, while Equation (A3) defines the function regulating the state evolution over time. It incorporates a dependence on the stochastic variable $w_t$ that can assume values in the set $\{-1, 0, 1\}$, according to the forward, stationary, and backward transition probabilities, $P^F(u_t)$, $P^S(u_t)$, and $P^B(u_t)$. The forward probability results from the linear combination of two basic theoretical functions, $P^A$ and $P^I$, representing the propensities of individuals towards analytic or intuitive reasoning. The propensities are described by Equations (A4) and (A5), where $\overline{u}_t = 10(1 - u_t)$ is introduced to resize the variable $u_t$ so that it belongs to the set $U = (0, 1)$, and to allow the propensities $P^A$ and $P^I$ laying in the interval $[0, 1]$. The propensity to the *intuitive* reasoning is shaped similarly to the one proposed by Moerland et al. in their experimental research [36]. $P^A$ and $P^I$ are combined using as coefficient the reasoning propensity $p_r$ of the individual, thus giving rise to the forward transition probability of Equation (A6). The stationary transition probability is assumed constant (Equation (A7)). Finally, the backward transition probability is obtained by difference from the previous two, as described by Equation (A8).

The parameters' setting for $\alpha$, $\beta$, $\delta$, $\gamma$, and the initial state $s_0$ included in the model equations are varied in the simulations and will be clarified in the different reports of the results. Other parameters are assumed equal for all the simulations and set to the values $a = 1$, $b = 7$, $c = 10$, $d = 0.1$, $k = 0.1$.

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
