# Peer review of "Decision Making in Networks: A Model of Awareness Raising"

_information, doi:10.3390/info14020072_

Round 1

Reviewer 1 Report

The paper presents an interesting math model in explaining a social behavior. The introduction provides some background on the concepts of interaction and awareness but it needs more transition on why it needs a mathematical model in the research. 

Author Response

Reviewer 1

The paper presents an interesting math model in explaining a social behavior. The introduction provides some background on the concepts of interaction and awareness but it needs more transition on why it needs a mathematical model in the research.

  1. We thank the reviewer for their suggestions. We modified Section 1. Introduction, trying to explain the utility to study these phenomena from a mathematical perspective. In particular, we added a paragraph and some references specifically devoted to explaining the meaning and importance of the use of mathematical models of awareness:

“The interest in developing such a mathematical model of awareness is manifold. From a theoretical point of view, it provides a reference for evaluating the impact of personal relations on awareness raising, thus allowing us to analyze it from both the individual and the social perspectives [3]. Moreover, it can apply to investigate the pros and cons of using artificial intelligence in human decision-making and identify new directions for integrating a prototypical consciousness into AI algorithms [4,5]. From this point of view, it is worth noting that the different individuals interacting with each other could be, in the most generic view, also non-human agents, AIs or robots for example [6]. Obviously, in these cases the concept of awareness has a different meaning, albeit retaining both its character of ‘slow accumulation’ of knowledge from past examples and of the generation of ‘contextual’ knowledge at a different hierarchical level with respect to step-by-step optimization.“

Reviewer 2 Report

paper is about mathematical model to quantify interpersonal traits. Comments are:

- title needs editig, present title conveys no meaning

- abstract must be rewritten, what is the need for thework, what is the problem with earlier work, what is the proposed idea, what is the objective, where is the datasource

- introduction must be edited, state the problem clearly, what are the limitations of extant works, what are the contributions, state the research hypothesis clearly

- If Markov model is already presented, what is the idea proposed here for quantification

- review must be made comprehensive, details about Markov models must be provided and also interpersonal aspects must be discussed in detail

- what is the application studied in the work and what is the model focussed must be clarified

- Give step by step working of the method, at each method elaborate the input size and output size and show a worked example with clear stepwise procedure for all readers to understand the work

- comparison is not comprehensive, use different statistical measures to evaluate the model with other works

- what are the limitations of the present work, what are the benefits of the present work must be clarified

- avoid typo in references please

Round 2

Reviewer 2 Report

Revision is undertaken by authors, still some comments needs focus

- please edit title to bring out what exactly is done. What new method is presented and so on. This looks more like a conference theme, please edit it.

- comparison can be improved with more statistical metrics.

Author Response

Dear Editor,

please find below the replies to the points raised by the reviewer. All changes described below are tracked in red or blue in the newly uploaded manuscript version.

  • 1. please edit title to bring out what exactly is done. What new method is presented and so on. This looks more like a conference theme, please edit it.
  • R1. A new title, more significant and explicative of the method used is proposed. The new title is: 

    Decision Making on Networks: a Model of Awareness Raising

  • 2. comparison can be improved with more statistical metrics.
  • R2. The statistical comparison between the model simulations reported in Figure 2 has been performed by evaluating the mean and variance of the agents' behaviors. The results are reported in the new Table 1 on page 10 of the manuscript.

We sincerely thank the Editor and the Reviewers for their efforts in reviewing our paper.

Sincerely yours,

Chiara Mocenni

(on behalf of all authors)